# FASTER CNNs WITH DIRECT SPARSE CONVOLUTIONS AND GUIDED PRUNING

**Jongsoo Park**[1]        **Sheng Li**[1]        **Wei Wen**[2]        **Ping Tak Peter Tang**[1]

**Hai Li**[3]                **Yiran Chen**[3]                **Pradeep Dubey**[1]

[1]Intel Labs, [2]Department of Electrical and Computing Engineering, University of Pittsburgh
[3]Department of Electrical and Computer Engineering, Duke University
[1]{jongsoo.park, sheng.r.li, peter.tang, pradeep.dubey}@intel.com,
[2]{wew57}@pitt.edu, [3]{yiran.chen, hai.li}@duke.edu

## ABSTRACT

Phenomenally successful in practical inference problems, convolutional neural networks (CNN) are widely deployed in mobile devices, data centers, and even supercomputers. The number of parameters needed in CNNs, however, are often large and undesirable. Consequently, various methods have been developed to prune a CNN once it is trained. Nevertheless, the resulting CNNs offer limited benefits. While pruning the fully connected layers reduces a CNN's size considerably, it does not improve inference speed noticeably as the compute heavy parts lie in convolutions. Pruning CNNs in a way that increase inference speed often imposes specific sparsity structures, thus limiting the achievable sparsity levels.

We present a method to realize simultaneously size economy and speed improvement while pruning CNNs. Paramount to our success is an efficient general sparse-with-dense matrix multiplication implementation that is applicable to convolution of feature maps with kernels of arbitrary sparsity patterns. Complementing this, we developed a performance model that predicts sweet spots of sparsity levels for different layers and on different computer architectures. Together, these two allow us to demonstrate 3.1–7.3× convolution speedups over dense convolution in AlexNet, on Intel Atom, Xeon, and Xeon Phi processors, spanning the spectrum from mobile devices to supercomputers.

## 1 INTRODUCTION

Due to the success of deep neural networks in a broad set of practical and even critical artificial intelligence tasks, they are now widely deployed in a spectrum of platforms: smart phones, autonomous cars, data center servers, and even supercomputers. While suitably designed and trained CNNs can be powerful, they are often large – requiring many parameters (e.g., the celebrated AlexNet (Krizhevsky et al., 2012) has 60 millions). That large neural network models incur cost in terms of memory, energy, and inference speed is easy to see.

This motivated a line of research (Han et al. (2015; 2016b); Guo et al. (2016); Denton et al. (2014), to name a few) that tries to prune the parameters after a CNN design is trained and proved useful. A common thread is to post-process a trained CNN. Post-processing may consist of retraining with sparsity inducing regularization or of approximating tensors of parameters via tensor factorization. These methods reduce the size of CNNs significantly while preserving inference accuracy. Nevertheless, the inference speed gains in pruned networks is not nearly as impressive as the size reduction. In this sense, the benefits of CNN pruning seem not fully realized.

While seemingly unintuitive, that the significantly pruned CNNs run not nearly as significantly faster can be easily explained. First, fully connected (`fc`) layers usually contain the bulk of the parameters

while convolutional (`conv`) layers consume the bulk of computation time. This property shows that reducing the size of just the `fc` layers will readily lead to meaningful reduction in size as in Han et al. (2016b); Guo et al. (2016); but little speed improvement.

The crux of speed improvement thus lie in actual fast convolution of sparse kernels with feature maps (not just floating-point operations reduction), which is a challenging problem. It is well known in the field of numerical linear algebra that the performance of sparse matrix operations is typically memory bandwidth bound. Direct application of the sparse matrix operations to compute the `conv` layers when the kernels are sparse will likely result in sub-optimal speed gains. This concern on low efficiency of sparse operations is also discussed in the design of GoogLeNet (Szegedy et al., 2015). We will term methods that work directly with sparse data structures "sparse methods." Alternative to sparse methods, "dense methods" gather data in a way that allow the actual convolution be performed by dense linear algebra functions such as GEMM. An example is found in (Lebedev & Lempitsky, 2015; Wen et al., 2016) which produces some group-wise sparsity patterns that facilitate the use of existing and highly tuned dense matrix computation library functions to perform the convolutions. However, imposing sparsity patterns limits the sparsity level that would otherwise be achievable had arbitrary patterns been allowed. We note that high compression in the `conv` layers are gaining importance as these layers consist a much larger percentage of parameters in recent networks such as GoogLeNet (Szegedy et al., 2015) and ResNet (He et al., 2015).

We view sparse methods differently. Convolutions in CNNs involve multiple channels and thus offer much higher data reuse than typical sparse matrix operations in scientific computing. Specifically, we present a highly efficient direct sparse convolution design formulated as sparse-matrix-dense-matrix multiplication with the dense matrix columns generated on-the-fly from a single column vector. In addition to being highly efficient, this sparse convolution design is friendly with convolution kernels with arbitrary sparsity patterns. We call this *element-wise sparsity* to distinguish it from *group-wise sparsity* mentioned previously. As shown later on, accepting element-wise sparsity significantly increases the achievable sparsity level.

Complementing our sparse convolution design, we formulate a performance model to elucidate when and how best to use sparse convolutions on different computer architectures and at different CNN layers. Our formulation follows the roofline model (Williams et al., 2009). In particular, our model suggests (correctly) that sparse convolution can improve inference speed even with a moderate sparsity level of around 70%. In addition, the model provides upper and lower bounds of sparsity levels that can contribute to speed improvements. Sparsity higher than the upper bound offer no further speed improvement; and sparsity lower than the lower bound can in fact slow down inference rather than accelerating it.

Combining the sparse convolution design with the performance model allows us to prune a CNN in a co-design manner, with our proposed new pruning algorithm—Guided Sparsity Learning (GSL). As illustrated later, we can adjust sparsity targets precisely at different layers so as to maximize inference speed, best preserve accuracy, and minimize a network's size. In some cases, particular layers are identified as best not pruned at all due to no potential speedups, leaving them unchanged gives other layers more room for gainful pruning in terms of size and speed.

Our paper makes the following contributions:

- A high performance sparse convolution design that takes advantage of arbitrary sparsity patterns and outperforms dense convolution even with a moderate sparsity.

- A general performance model that (1) projects speedups over dense convolutions on varying level/types of sparsity and different computing platforms and (2) provides training guidelines for precisely targeting layers and sparsity ranges with potential to accelerate inference.

- Guided Sparsity Learning (GSL), the first pruning algorithm fusing the awareness of speedup potential into sparsity learning; and its application to AlexNet and GoogLeNet. In particular, in GoogLeNet, we prune out more than 80% of parameters of all $5\times5/3\times3$ `conv` layers and `fc` layers with no accuracy drop.

- An optimized sparse convolution implementation (http://github.com/IntelLabs/SkimCaffe) that provides $7.3\times$, $3.4\times$, $3.1\times$ speedups of convolution layers in AlexNet over dense methods on Intel Atom, Xeon, and Knights Landing processors, respectively, with no accuracy drop. In particular, this paper is one of the first evaluations of Xeon Phi processors on deep learning algorithms.

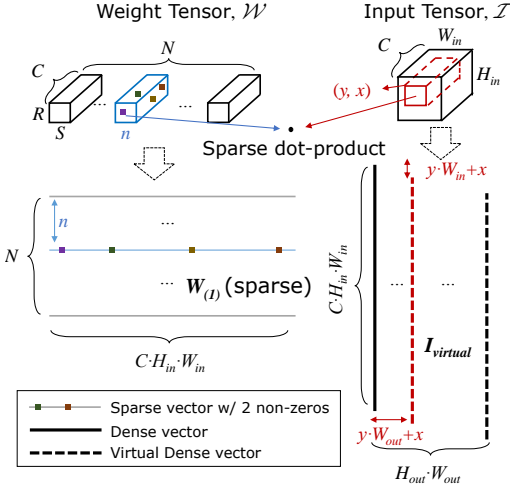

Figure 1: Conceptual view of the direct sparse convolution algorithm. Computation of output value at $(y, x)$th position of $n$th output channel is highlighted.

```
for each output channel n {
 for j in [W.rowptr[n], W.rowptr[n+1]) {
  off = W.colidx[j]; coeff = W.value[j]
  for (int y = 0; y < H_OUT; ++y) {
   for (int x = 0; x < W_OUT; ++x) {
    out[n][y][x] += coeff*in[off+f(0,y,x)]
   }
  }
 }
}
```

Figure 2: Sparse convolution pseudo code. Matrix **W** has *compressed sparse row* (CSR) format, where `rowptr[n]` points to the first non-zero weight of $n$th output channel. For the $j$th non-zero weight at $(n, c, r, s)$, `W.colidx[j]` contains the offset to $(c, r, s)$th element of tensor `in`, which is pre-computed by layout function as $f(c, r, s)$. If `in` has CHW format, $f(c, r, s) = (cH_{in} + r)W_{in} + s$. The "virtual" dense matrix is formed on-the-fly by shifting `in` by $(0, y, x)$.

The rest of the paper is organized as follows. Section 2 presents the details of our sparse convolution design, formulation of the performance model, the Guided Sparsity Learning (GSL) pruning algorithm, and how they are combined to prune and accelerate CNNs. Section 3 demonstrates the effectiveness of these developments on AlexNet and GoogLeNet on a variety of platforms. Section 4 discusses related works and review the state of the art. Section 5 concludes and outlines a few next-steps.

## 2 GOING FASTER WITH DIRECT SPARSE CONVOLUTION, PERFORMANCE MODEL, AND GUIDED PRUNING

As explained previously, prunning CNN models does not yet benefit inference speed as much as model size reduction. This section first presents our efficient direct sparse convolution design that remedies this situation significantly. We then develop a performance model that projects speedup over different sparsity levels and on different processor architectures. The model guides our speedup-aware training method, Guided Sparstiy Learning (GSL).

### 2.1 DIRECT SPARSE CONVOLUTION

A sparse convolution for the all output positions across all output channels can be eventually considered as a *virtual* sparse-matrix-dense-matrix multiplication (SpMDM), as described in the following. Consider a bank of $N$ filters each with size $R \times S$ against an $H_{in} \times W_{in}$ feature with $C$ input channels. We denote the filter bank as a 4-mode tensor $\mathcal{W}$ with size $N \times C \times R \times S$, the input feature as a 3-mode tensor $I$ with size $C \times H_{in} \times W_{in}$, and the output feature as a 3-mode tensor $O$ with size $N \times H_{out} \times W_{out}$. The output value at $(y, x)$th position of $n$th output channel is computed by

$$O(n, y, x) = \sum_{c=0}^{C-1} \sum_{r=0}^{R-1} \sum_{s=0}^{S-1} \mathcal{W}(n, c, r, s) I(c, y + r, x + s), \qquad (1)$$

which is a dot-product of two 3D tensors as shown in Figure 1. This can be treated as a vector dot-product by: first vectorizing the 3D subtensor of $\mathcal{W}$ corresponding to the $n$th output channel, then vectorizing $I$ (denoted as $vec(I)$), and finally stretching the first vector to match the dimension of two vectors. When $\mathcal{W}$ is sparse, then this vector dot-product becomes a sparse-vector-dense-vector dot-product. Consider flattening dimensions except the first one of $\mathcal{W}$ into a sparse matrix $\mathbf{W}_{(1)}$ (i.e. mode-1 matricization of $\mathcal{W}$ as in Kolda & Bader (2009)), with its row vectors stretched to match the dimension of $vec(I)$. $O(n, y, x)$ is then the dot-product between the $n$th row of $\mathbf{W}_{(1)}$ and $vec(I)$. Subsequently, the values at the same given $(y, x)$th position of all $N$ output channels can be computed

collectively as a sparse-matrix-dense-vector multiplication (SpMV):

$$\mathbf{O}_{(1)}\left(:,yW_{out}+x\right) = \mathbf{W}_{(1)} \cdot vec\left(I_{y,x}\right), \tag{2}$$

where $I_{y,x}$ denotes the tensor $I$ with its last two dimensions shifted by $(y,x)$. The values at different output positions can be computed as a sparse-matrix-dense-matrix multiplication (SpMDM), where the columns of the dense matrix are actually the same vector $vec(I)$ but with different offsets. In order to save bandwidth usage, we operate with a *virtual* dense matrix, $\mathbf{I_{virtual}}$, where its columns are generated on the fly by adjusting indices through which we access $vec(I)$.

Using the virtual dense matrix essentially skips the lowering step used in standard frameworks such Caffe, and, therefore, we call our method *direct sparse convolution*, to distinguish it from sparse convolution with lowering such as the method used in Liu et al. (2015). The lowering approach replicates the input feature multiple times, significantly reducing arithmetic intensity. The lowering process has demonstrated overhead for dense convolution as in Hadjis et al. (2015); Chintala (2015), and is particularly problematic for sparse convolution with intensity already lower than its dense counter part. Figure 3 demonstrates the advantage of our direct sparse convolution, using the performance model that will be developed Section 2.2, where our direct sparse convolution significantly outperforms lowering-based methods at a high level of sparsity.

Even though direct sparse convolution may seem conceptually more complicated than the usual SpMDM or the lowering-based methods, it can be concisely expressed in the pseudo code shown in Figure 2. To decouple from a specific layout of tensor $I$, the pseudo code uses layout function $f$ such that $f(c,y,x)$ maps to the offset corresponding to $(c,y,x)$th element of $I$ (we assume $f(c,y+r,x+s) = f(c,y,x) + f(0,r,s)$). For example, in CHW layout, $f(c,y,x) = (c \cdot H_{in} + y)W_{in} + x$.

In convolutional layers in CNN, an input channel is reused against multiple output channels and vice versa, and there is also ample reuse out of an input channel due to overlapping between dot-products, especially for a large filter size and a small stride. Therefore, the arithmetic intensity of sparse convolution can be significantly higher than typical sparse-matrix computations such as SpMV, thus leading to high compute efficiency. Our optimized implementation[1] fully takes advantage of the reuse, applying loop tiling to both input and output channels, with column blocking (Buluç et al., 2009) applied to the weight sparse matrix. SIMDification and register blocking optimizations are also applied to the y and x loops in the pseudo code. Non-contiguous indirect access (i.e. gather) is another overhead of typical sparse-matrix computations. However, as shown in our pseudo code, the values read from colidx and value arrays of a sparse matrix are reused $H_{out} \cdot W_{out}$ times. The access to tensor in is also contiguous as long as the tensor's elements with contiguous $x$ or $y$ values are stored contiguously, which is a common case as in the CHW format.

## 2.2 ANALYTICAL PERFORMANCE MODELING: PROJECTIONS ON SPARSE CONVOLUTION SPEEDUP AND GUIDELINES ON USEFUL SPARSITY RANGE

The performance of sparse convolution depends highly on the sparsity level of the weight tensor. This section develops a performance model to determine the appropriate target sparsity range for pruning and to project theoretical speedup for any given sparsity, using the roofline model (Williams et al., 2009).

We denote the floating-point operations required in a convolution as $C$ (in FLOP), the size of input and output activation tensors as $S_A$ (in Bytes), and the size of weight tensor as $S_W$, all without considering sparsity. We denote the density of non-zero in filters as $x$ (the lower the $x$, the higher the sparsity of weight tensor), the compute capability of processor as $F$ (in FLOP/s), and the memory bandwidth as $B$ (in B/s). With these parameters, the time for dense convolution ($t_{dense}$), the time for sparse convolution bound by compute ($t_{sparse\_compute}$) and by bandwidth ($t_{sparse\_bw}$), and theoretical speedup can be modeled as follows (we assume dense convolution is not bandwidth bound):

$$t_{dense} = \frac{C}{F}, \quad t_{sparse\_compute} = \frac{\alpha x C}{F}, \quad t_{sparse\_bw} = \frac{S_A + \beta x S_W}{B}, \quad \text{speedup} = \frac{t_{dense}}{max(t_{sparse\_compute}, t_{sparse\_bw})},$$

where $\alpha$ and $\beta$ denote the compute and storage overheads of sparse representations, respectively. We observe $\alpha \sim 3$ on a Xeon E5-2697 v4 processor, and $\beta$ is typically 2 (in compressed sparse row representation, we need 4B column index for each 4B single-precision floating point non-zero value).

---

[1] https://github.com/IntelLabs/SkimCaffe/blob/intel_scnn/include/caffe/util/sconv.hpp

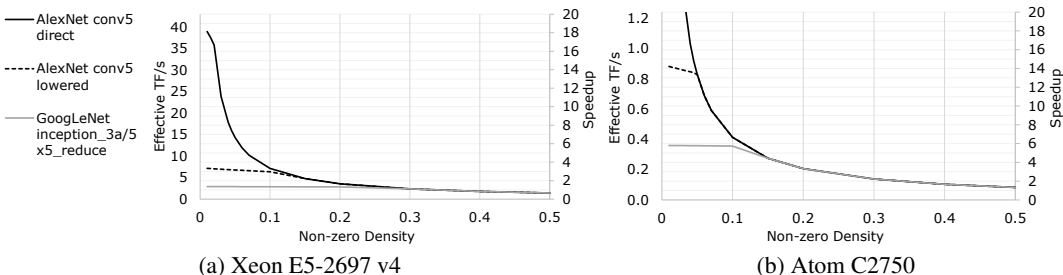

(a) Xeon E5-2697 v4 (b) Atom C2750

Figure 3: Projected performance of sparse convolution and its speedup over dense convolution for a Xeon processor and an Atom processor. `conv5 direct`: direct spares convolution, `conv5 lowered`: sparse convolution on tensors lowered to matrices. We use the processors' achievable FLOP/S and memory bandwidth shown in Table 1 and the compute overhead of sparse convolution measured in Section 3.2.

This analytical model is visualized in Figure 3. Here, we define *effective FLOP/s* with respect to the number of floating-point operations that would have been performed by dense convolutions including the ones for zero weights (i.e. effective FLOP/s = $C/t_{sparse}$). With a moderate sparsity (e.g., $x = 0.2$), convolution is likely to be compute bound, and hence effective FLOP/s rapidly increases as $x$ decreases. For `conv5` in AlexNet with $x \in (0.05, 0.15)$, a typical sparsity range without accuracy loss, direct sparse convolution can achieve 2–7× and 4–14× speedup on Xeon and Atom platforms, respectively, as shown in Figure 3 and will be validated in Section 3.2.

However, decreasing arithmetic intensity further with lowering $x$ eventually makes the performance bandwidth bound. Thus, *there is an upper bound of useful sparsity, and a sparsity higher than it does not provide additional speedup, while only making training more challenging to preserve accuracy.* This upper bound can be found by solving for $x$ such that $t_{sparse\_compute} = t_{sparse\_bw}$ (e.g., the upper bound sparsity for `conv5` of AlexNet on the Xeon is $x \sim 0.02$). This analysis can be applied to various computing platforms including CPUs and GPUs because the model captures the essential platform-dependent characteristic, the ratio of bandwidth compute capability to memory bandwidth ($F/B$). When the compute to bandwidth ratio is lower as in a platform like Atom, the performance will be less quickly bandwidth bound. For example, the lower bound of useful sparsity for `conv5` of AlexNet is $x \sim 0.01$ on Atom C2750, which is smaller than that of Xeon. The speedup to sparsity relation also varies over layers. For example, since 1×1 convolutions in GoogLeNet has low arithmetic intensity to begin with, its performance quickly becomes bandwidth bound at lower sparsity (or higher $x$).

The compute overhead, α, depends on the quality of sparse convolution implementation and on the target processor architecture. Since $t_{sparse\_compute} > t_{dense}$ for $x > 1/\alpha$, *there is a lower bound of useful sparsity such that, with a sparsity lower than that, sparse convolution becomes slower than dense convolution.* The previous section described our sparse convolution implementation that achieves α=3 (since α is the compute overhead, lower is better) on the Xeon instead of α=100 as conjectured by Szegedy et al. (2015)[2].

### 2.3 GUIDED SPARSITY LEARNING (GSL)

The upper and lower bounds on useful sparsity can provide important insights for training/pruning. The model can tell that sparse convolution is not useful for certain layers, in which case we can skip pruning of those layers to provide more room for sparsity in the other layers. For example, layers like the first layer in AlexNet and GoogLeNet may not provide enough sparsity regardless of the amount of regularization applied as long as the original inference accuracy is to be preserved. A layer may be already bandwidth bound even before pruning like 1×1 convolution layers in GoogLeNet as shown by `inception_4a/5x5_reduce` layer in Figure 3.

---

[2]The compute overhead of α=3 primarily comes from that access to input tensor is not aligned at cache line boundaries. Recent Xeon processors can execute 1 unaligned SIMD load per cycle, which is not enough to sustain 2 SIMD fused multiply-add operations per cycle. In addition to this 2× overhead, when $W_{out}$ is not a multiple of SIMD width (8 for Xeon E5-2697 v4), we do not fully utilize the SIMD registers. Since Atom processors do not execute multiple SIMD floating operations per cycle anyway, and because its SIMD width is narrower as 4, its compute overhead is smaller as 1.2 as will be shown in Section 3.2.

Guided Sparsity Learning (GSL), our new pruning algorithm, is inspired by the insights and our performance model. GSL is the *first* to fuse the awareness of speedup potential into sparsity learning. GSL is a generic algorithm and accepts different regularization methods for pruning. When GSL is used with *element-wise* regularization for pruning, thus denoted as Guided Element-wise Sparsity Learning (GESL), it learns the element-wise sparsity of layers where the model predicts speedups. GSL can also be used with regularization methods that are more complicated than basic ridge and lasso regularization. For example, GSL can be combined with dynamic network surgery (Guo et al., 2016), as will be shown in Section 3.1

| Algorithm: Guided Sparsity Learning (GSL) |
|---|
| **Input:** Pruning layer set ($S$), performance model ($M$) |
| **Initialize** Project speedup for each layer $L$ in $S$ using $M$; Exclude all $L$s without speedup potential from $S$ |
| **Repeat** Train the whole neural network while actively pruning only $L$s in $S$ |
| Project speedup of $L$s in $S$, using their current sparsity and $M$, periodically |
| Periodically, for each $L$ in $S$ do: |
| if (sparsity $\geq$ upper bound of the useful sparsity range), stop pruning $L$ |
| if (stabilized sparsity $\leq$ lower bound), stop pruning $L$ & restore its original dense weights |
| /∗ Stop pruning $L$ is to give other $L$s better chance to prune further and achieve better accuracy ∗/ |
| **Until** Maximum iterations or convergence reached |

Although GSL as described above aims primarily at inference speed, GSL can balance the implications of pruning on inference speed, accuracy, and model size. To do this, optional constraints can be given to GSL to prioritize pruning of different layers in the network. For example, by using different regularization strengths on `conv` and `fc`, we can tune the priorities on speed and model size.

## 3 EXPERIMENTS

Our sparse CNN design is evaluated on three platforms shown in Table 1. Intel C2750 (`Atom`) represents resource-constrained mobile platforms or micro servers optimized for energy efficiency. Xeon E5-2697 v4 (`BDW`) represents data-center servers. Xeon Phi 7250 (`KNL`) is designed for high-performance computing, but its next version, Knights Mill, will specifically target machine learning. Our sparse CNN is implemented as an extension of Caffe deep learning framework (Jia et al., 2014) and is at https://github.com/IntelLabs/SkimCaffe. We use Intel compiler version 17.0.0 and use all cores available. The `SGEMM` performance and achievable memory bandwidth listed are measured with Intel MKL version 2017 and `STREAM` benchmark (McCalpin), respectively.

We train with the ImageNet ILSVRC-2012 dataset (Deng et al., 2009), starting from the pre-trained Caffe reference model (a slight variation but we call it AlexNet for simplicity) and GoogLeNet model from the Caffe model zoo. Since we find it is easy to have high sparsity with smaller networks and datasets like LeNet and CIFAR regardless of pruning method, we do not present their results. Our training process is based on the method described in Wen et al. (2016) with the following differences. We look for element-wise sparsity with lasso instead of group lasso, and guide the training process to target the layers and range of sparsity where we see speedup potential. We have explored various solver methods and learning rate schedules, but found that they do not significantly affect the eventual accuracy and sparsity, once hyper-parameters are tuned for the respective settings. In general, the pruning step no longer improves after 450K and 900K mini-batch iterations for AlexNet and GoogLeNet, respectively. The re-training step saturates around 150K and 300K mini-batch iterations. To see trade-offs among accuracy, speed, and model size, we try various weight decays ranging from 1e-5 to 1e-3, and, for AlexNet, decay multipliers for `fc` layer ranging from 1e-2 to 1. We find that the

Table 1: Evaluated Platforms

|  | Atom C2750 (`Atom`) | Xeon E5-2697 v4 (`BDW`) | Xeon Phi 7250 (`KNL`) |
|---|---|---|---|
| Socket×core×SP-SIMD | 1×8×4 | 1×18×4 | 1×68×16 |
| Clock (GHz) | 2.4 | 2.3 | 1.4 |
| SGEMM GFLOP/s | 62 | 2,150 | 4,540 |
| Achievable bandwidth (GB/s) | 15 | 122 | 480 |

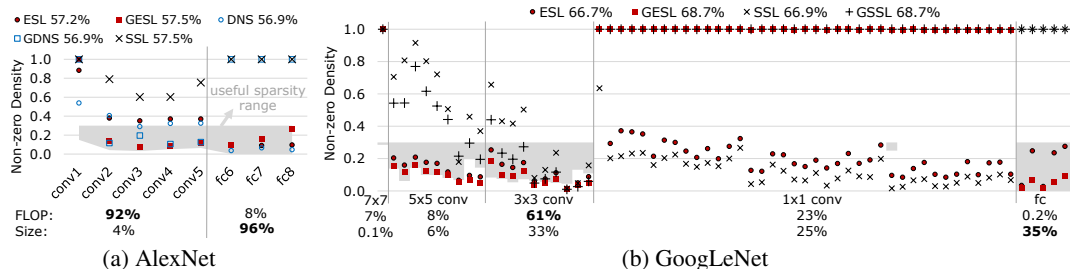

(a) AlexNet                                          (b) GoogLeNet

Figure 4: Layer-by-layer sparsity from element-wise sparsity learning (ESL), guided ESL, dynamic network surgery (DNS), guided DNS, and structured sparsity learning (SSL). The accuracies shown in percentage are top-1 accuracy measured with the ImageNet test set. The original AlexNet and GoogLeNet top-1 accuracies are 57.4% and 68.9%. DNS and SSL AlexNet results are from Guo et al. (2016) and Wen et al. (2016). GDNS AlxeNet and SSL GoogLeNet results are our own but with the same code used in their papers. The shaded area marks the useful sparsity range predicted by our model for `BDW`. No shaded area means sparse convolution is not useful for the layer regardless of sparsity. In GoogLeNet, we organize layers by their types, and, within each layer type, layers are ordered from the earlier layers in forward propagation.

starting learning rate of 1e-3 and weight decay of 5e-5 in general gives a high sparsity with minimal accuracy drop. We reduce the learning rate by $10\times$ for re-training step.

## 3.1 GUIDED TRAINING RESULTS

Figure 4 shows the effectiveness of our guided pruning and compares the level of element-wise and group-wise sparsity we can obtain. *We should look at layer-by-layer because the speedup over dense convolution does not have a simple linear relation with sparsity as shown by our model, and, therefore, the overall FLOP reduction does not necessarily closely correlate with the real speedup.* In AlexNet, using the same element-wise regularization factor across all layers (element-wise sparsity learning, ESL) provides non-zero densities around 0.4 for `conv2-5`. This is fine sparsity when the primary goal is reducing model size, but not high enough for speeding-up inference. Therefore, guided ESL (GESL) reduces the regularization factor of `fc` layers (as they have fewer FLOPS) and avoid pruning `conv1` entirely (as its sparsity is too low for any potential speedups with more regularization). This leads to less than 0.2 non-zero density for `conv2-5`, the range where we can get speedups from sparse convolution. Similarly, applying GSL to dynamic network surgery (DNS), a recent proposal to obtain high sparsity, as Guided DNS (GDNS), we can see that GSL effectively improve the obtained sparsity for accelerating inference by de-prioritizing `conv1` and `fc` layers (we go further to not prune `fc` layers at all to see how much sparsity DNS can provide in `conv` layers)[3].

Structured sparsity learning (SSL) provides group-wise sparsity, for which we can use dense methods, but its sparsity is lower because of constrained forms of sparsity. According to our model, SSL performs better when $x_g < (\alpha/\alpha_g)x$, where $x$ and $x_g$ are non-zero density of ESL and SSL, and $\alpha$ and $\alpha_g$ are the compute overhead of ESL and SSL, respectively. Even if we use an *ideal* 100% efficiency for SSL ($\alpha_g = 1$)[4] and the *measured* overhead $\alpha = 3$ for ESL, $x_g$ shown in Figure 4(a) is not small enough to outperform GESL. Note that our guiding principles are already applied to SSL, where `conv1` and `fc` layers are not pruned. In short, sparsity SSL can currently obtain is too low to outperform once compared with an optimized sparse convolution design for element-wise sparsity such as ours. This motivates further investigation of pruning methods for higher group-wise sparsity.

GoogLeNet has many $1\times1$ convolutions, for which sparse convolution does not provide speedups due to their low arithmetic intensity, as our model predicts. As shown in Figure 4(a), GESL successfully discovers this and avoids pruning the $1\times1$ convolutions for higher sparsity in $3\times3$ and $5\times5$ convolutions, where our model projects speedup, and, more importantly, almost recovers the original accuracy. For $1\times1$ convolutions, group-wise sparsity implemented in SSL reduces to

---

[3]Although not shown in Figure 4(b), we also apply DNS and GDNS to GoogLeNet. Compared to DNS, GDNS on GoogLeNet successfully reduces non-zero density by $1.4\times$ on average in layers with speedup potential by prioritizing these layers for pruning.

[4]Note that certain kinds of group-wise sparsity like "column-wise sparsity" defined in Wen et al. (2016) need lowering, which can be considerable overhead, making it hard to approach the ideal efficiency.

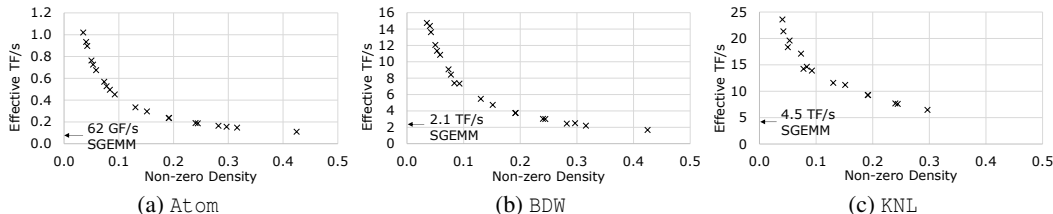

Figure 5: Performance of `conv2-5` layers of AlexNet with varying sparsity on Atom C2750 (a), Xeon E5-2697 v4 (b), and Xeon Phi 7250 (c). `SGEMM` performance of each platform serves as a proxy to the performance of dense convolutions.

element-wise sparsity[5] , and dense methods can no longer be used. We believe that SSL provides higher sparsity for $1 \times 1$ convolutions than ESL because SSL does not prune `fc` layers, providing more room to prune other layers[6]. For larger convolutions that contribute to the bulk of FLOPs, ESL provides significantly higher sparsity than SSL; most of the larger convolution layers have non-zero density less than 0.2, where sparse convolutions can provide speedups. It is interesting to note that ESL, GESL, and SSL all achieve very high sparsity of non-zero density less than 1.5% in layers like `inception_4e/3x3`. This may indicate that the $3 \times 3$ path is unnecessary in that inception module.

## 3.2 LAYER-BY-LAYER SPARSE CONVOLUTION RESULTS

Figure 5 shows layer-wise performance of our direct sparse convolution design with the useful high sparsity obtained from GESL. We evaluate with the sparse matrices from multiple pruned AlexNet models with up to 3% top-1 accuracy drop. Since the performance of sparse matrix operations highly depends on specific sparsity pattern, it is important not to evaluate with random sparse matrices. We use `SGEMM` as a proxy of dense convolution performance to quantify layer-wise speedups of direct sparse convolution. `SGEMM` is a good proxy because it has a long history of extensive optimizations, and it allows us not to depend on the quality of a specific dense convolution implementation[7]. We use batch sizes of 32, 144, and 272 for `Atom`, `BDW`, and `KNL`, multiples of the number of hardware threads in respective platforms.

`BDW` achieves $3.4\times$ speedup with non-zero density $x = 0.09$, the sparsity similar to those of `conv2-5` with no accuracy drop. The actual TF/s (as opposed to effective TF/s that also counts FLOPs for zeros) is 0.76 when sparse convolution is sufficiently compute bound (e.g., $x > 0.4$). This performance corresponds to about a third of `SGEMM`, from which we can derive the compute overhead of sparse convolution $\alpha$ as 3. As explained in Section 2.2, this leads to the lower-bound of sparsity to get speedups at $x = 0.3$, which matches with Figure 5(b). `Atom` with a higher bandwidth to flop ratio achieves higher $7.3\times$ speedup at $x = 0.09$. The actual GF/s is 51 when $x > 0.4$, which is $1.2\times$ lower than `SGEMM` performance (i.e. $\alpha = 1.2$). Note that the performance projection for `conv5` in Figure 3 using $\alpha$s derived here resembles the measured performance in Figure 5 (`conv2-5` share similar performance characteristics). `KNL` achieves impressive 13.9 effective TF/s at $x = 0.09$ ($3.1\times$ over `SGEMM`).

## 4 RELATED WORK

Recent researches have achieved great success on reducing model size and accelerating inference of CNNs while maintaining accuracy, exploring a large design space as shown in Table 2. Regularization-based and factorization-based approaches are the two main camps. Regularization-based approaches

---

[5]This is because filter coefficients for a given input and output channel pair is also a type of group that SSL is looking for.

[6]However, no pruning in `fc` layers is not an inherent limitation of SSL. This is just because Wen et al. (2016) focus on `conv` layer, and we follow the same approach to see maximum sparsity that SSL and GSSL can get in `conv` layers.

[7]This is the same reason for this paper to focus on layer-wise performance instead of overall end-to-end speedup. As the baseline for overall end-to-end speedup may be relative to a baseline whose efficiency is suboptimal with performance bottlenecks in other parts/layers of the code. For more scientific comparison among different CNN speedup techniques, we recommend using dense matrix multiplication (GEMM) FLOP/s of the evaluated platform as the baseline, because many platforms readily have vendor-provided extensively-optimized GEMM implementations which can be a proxy of highly-optimized dense CNN implementation. This also aligns with a long-accepted standard practice in high performance computing community.

| | A: Lebedev & Lempitsky (2015)[*], Wen et al. (2016)[*] | B: Han et al. (2015), Han et al. (2016b), Liu et al. (2015)[*], Guo et al. (2016), GESL[*] | C: Denton et al. (2014), Jaderberg et al. (2014), Lebedev et al. (2015), Zhang et al. (2015), Kim et al. (2016), Ioannou et al. (2016), Tai et al. (2016), Denil et al. (2013) |
|---|---|---|---|
| Pruning | Regularization | | Factorization |
| | Group-wise | Element-wise | |
| Computing | Dense | Sparse | Dense |

Table 2: Design space of techniques in reducing model size and accelerating inference, categorized as 3 groups. The footer rows of the table specify the two pillars of design space: pruning methods (how the sparsity is obtained during training) and computing methods (how the the obtained sparsity during inference). For techniques using regularization for pruning, [*] denotes those focusing more on `conv` layers than on `fc` layers.

use a separate training step to discover and prune redundant parameters in a pre-trained model using various regularizations, including ridge (Han et al., 2015; 2016b), lasso, and group lasso (Liu et al., 2015; Wen et al., 2016), combined with thresholding. Factorization-based approaches use low-rank decomposition and can quickly produce compressed models without additional pruning steps. Both approaches can use a fine-tuning step to recover accuracy loss caused by model pruning.

Researches focusing on fully connected layers (Han et al., 2015; 2016b; Denil et al., 2013) have achieved 10–50× model size reduction for networks such as AlexNet (Krizhevsky et al., 2012). However, they achieved marginal inferencing speedup because fully connected layers usually account for less than 10% of total computation in modern CNNs. Researches in groups A and C shown in Table 2 aim at speeding up inference by focusing more on convolution layers, with most of them relying on dense methods for computing convolution. While factorization-based approaches (group C) obtain smaller models in dense format naturally, regularization-based approaches (group A) need group regularization to impose group-wise sparsity. Although Liu et al. (2015) explore sparse methods in computing convolution layers, their approach involves lowering overhead and uses hard-coding non-zeros in sparse matrix with full unrolling that leads to a large instruction footprint.

While our direct sparse convolution is demonstrated to achieve high speed up on convolution when having enough sparsity, factorization-based approaches can complement. This is because the inherent sparsity in the first few convolution layers can be not high enough, while factorization-based approaches can achieve speedups there. Liu et al. (2015) also show that factorization and regularization-based approaches can be combined.

Winograd (Lavin & Gray, 2015) and FFT based algorithms (Vasilache et al., 2015) also aim to speedup convolution. While being orthogonal, these techniques can have synergies with our direct sparse convolution. For example, FFT based convolutions are more effective for large filters that usually reside in the first few layers where sparsity is low. While this paper focuses on convolution layer performance, our technical report (Park et al., 2016) also considers optimizations for fully connected layers, and sparsity in activations, which is also discussed in Han et al. (2016a).

## 5 CONCLUSIONS

Powerful CNNs are often quite compute demanding. Pruning as a post-processing step has been effective in drastically reducing the model size while boosting inference speed moderately. We aim to more fully realize the potential performance benefits due to the reduced FLOP counts resulting from pruned convolution kernels. By combining our high-performance direct sparse convolution method with a performance model, we developed a guided approach that prunes CNNs in a co-design fashion for different computer architectures and on different layers of a CNN in question. In particular, we demonstrated 3.1–7.3× convolution speedups in AlexNet on a variety of platforms, all in comparison to extensively-optimized dense linear algebra operations.

Looking ahead, as this paper shows that pruning can boost inference speed significantly in additional to reducing model size, further techniques in pruning should be explored. While our direct sparse convolution algorithm is successful, our performance model also reveals that sparse convolution cannot speedup all convolution layers, as seen from 1×1 convolutions in GoogLeNet. We plan to expand our performance model to cover other FLOP-reduction methods such as FFT, Winograd, and

tensor factorization, so that we can make informed decisions to choose the best performing method for each layer and the training process can be guided accordingly.

ACKNOWLEDGEMENT

We would like to thank Yiwen Guo, Anbang Yao, and Yurong Chen for sharing the dynamic network surgery source code and their insights. We would also like to thank Nitish Shirish Keskar for his recommendations on hyper-parameter settings.

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
