# Peer review of "Faster CNNs with Direct Sparse Convolutions and Guided Pruning"

_ICLR 2017 — accepted_

[Official Review · AnonReviewer1 · rating 6 · confidence 3 · 16 Dec 2016]
**Impressive empirical results; minimal research contribution**

The authors provide a well engineered solution to exploiting sparsity in convolutional layers of a deep network by recasting it as sparse matrix-vector multiplication. This leads to very nice speedups and the analysis of when this is possible is also useful for practitioners. My main concern with this paper is that the "research" aspect of it seems rather minimal, and it's mostly about performance engineering and comparisons. It is upto the area chairs to decide how well such a paper fits in at ICLR.

[Official Review · AnonReviewer2 · rating 6 · confidence 3 · 16 Dec 2016 (modified: 19 Dec 2016)]

This paper tackles the problem of compressing trained convnets with the goal of reducing memory overhead and speeding up the forward pass. As I understand it, the main contribution of this work is to develop fast convolution routines for sparse conv weights int he case of general sparsity (as compared with structured sparsity). They evaluate their method on both AlexNet and GoogLeNet as well as on various platforms. The authors make code available online. The paper is well written and does a good job of putting this work in the context of past model reduction techniques.

My main request of the authors would be to provide a concise summary of the speedup/memory gains achievable with this new work compared with previously published work. The authors do show the various sparsity level obtained with various methods of pruning but it is unclear to me how to translate the information given in the paper into an understanding of gains relative to other methods.

[Official Review · AnonReviewer3 · rating 7 · confidence 3 · 17 Dec 2016]

The paper details an implementation of sparse-full convolutions and a model to work out the potential speed-up of various sparsity levels for CNNs.

The first contribution is more about engineering, but the authors make the source code available which is greatly appreciated.

The second contribution is perhaps more interesting, as so far pruning methods have focused on saving memory, with very modest speed gains. Imbuing knowledge of running speed into a pruning algorithm seems like the proper way to tackle this problem. The authors are very methodical in how they build the model and evaluate it very thoroughly.

It seems that the same idea could be used not just for pruning existing models, but also when building new architectures: selecting layers and their parameters as to achieve an optimal throughput rate. This could make for a nice direction for future work.

One point that is missing is some discussion of how transferable the performance model is to GPUs. This would make the technique easier to adopt broadly.

Other areas for improvement: The points in Figure 4 are hard to distinguish (e.g. small red circle vs. small red square), and overall the figure could be made bigger; specifying whether the "base learning rate" in Section 3 is the start or end rate of the annealing schedule; typos: "punning" (p.4), "spares" (p.5).

[Final Decision · Program Chairs · 06 Feb 2017]
**ICLR committee final decision**

While the core ideas explored in this paper are quite limited in algorithmic novelty (e.g., the direct sparse convolutions), the reviewers largely feel that the paper is well written, experiments are carefully done on multiple architectures and system issues are discussed in-depth. Given the interest in the ICLR community around performance characterization and acceleration of CNNs in particular, this paper offers an interesting perspective.